# Identification of a Novel Model for Predicting the Prognosis and Immune Response Based on Genes Related to Cuproptosis and Ferroptosis in Ovarian Cancer

**DOI:** 10.3390/cancers15030579

**Published:** 2023-01-18

**Authors:** Ying Li, Tian Fang, Wanying Shan, Qinglei Gao

**Affiliations:** 1Key Laboratory of the Ministry of Education, Cancer Biology Research Center, Tongji Hospital, Tongji Medical College, Huazhong University of Science and Technology, Wuhan 430030, China; 2Department of Obstetrics and Gynecology, Tongji Hospital, Tongji Medical College, Huazhong University of Science and Technology, Wuhan 430030, China

**Keywords:** cuproptosis, ferroptosis, ovarian cancer, prognosis, immune infiltration, immunotherapy

## Abstract

**Simple Summary:**

Ovarian cancer is highly malignant with a poor prognosis, and there is still a lack of effective treatment methods. The exploration of modulating cell death processes has facilitated cancer treatment. Cuproptosis and ferroptosis are two novel forms of dying, and we sought to explore new biomarkers associated with them to guide the treatment of ovarian cancer. Our study established specific molecular types based on 39 genes related to cuproptosis and ferroptosis. And we systematically evaluated the differences in prognosis, drugs, and immunotherapy response between different subtypes of ovarian cancer. Molecular subtypes and risk model showed superior prognosis prediction and immune response prediction capabilities, which can provide a reference for personalized treatment of ovarian cancer.

**Abstract:**

(1) Background: Ovarian cancer (OV) presents a high degree of malignancy and a poor prognosis. Cell death is necessary to maintain tissue function and morphology. Cuproptosis and ferroptosis are two novel forms of death, and we look forward to finding their relationship with OV and providing guidance for treatment. (2) Methods: We derived information about OV from public databases. Based on cuproptosis-related and ferroptosis-related genes, a risk model was successfully constructed, and exceptional subtypes were identified. Next, various methods are applied to assess prognostic value and treatment sensitivity. Besides, the comprehensive analysis of the tumor environment, together with immune cell infiltration, immune function status, immune checkpoint, and human HLA genes, is expected to grant assistance for the prognosis and treatment of OV. (3) Results: Specific molecular subtypes and models possessed excellent potential to predict prognosis. Immune infiltration abundance varied between groups. The susceptibility of individuals to different chemotherapy drugs and immunotherapies could be predicted based on specific groups. (4) Conclusions: Our molecular subtypes and risk model, with strong immune prediction and prognostic prediction capabilities, are committed to guiding ovarian cancer treatment.

## 1. Introduction

Ovarian cancer, a malignant tumor of reproductive organs, ranks third in incidence after cervical and uterine cancer. A total of 70% of OV patients are advanced at the presentation, and resistance to chemotherapy drugs makes treatment options limited and ineffective. Unfortunately, effective screening and early diagnostic measures are still lacking. Therefore, there is an urgent need to explore pathogenesis and more effective treatments [1].

The exploration of modulating cell death processes has recently facilitated cancer treatment, mainly including apoptosis [2], necroptosis [3], and pyroptosis [4]. Ferroptosis is an iron-mediated death characterized by excessive accumulation of lipid peroxides and reactive oxygen species (ROS), ultimately leading to cell membrane damage [5,6]. As of now, ferroptosis is regarded as a potential therapeutic target for OV. The proliferative capacity of ovarian cancer cells is significantly inhibited by the activation of ferroptosis. Of significance, the report has revealed that PARP inhibitors (PARPi) can also promote ferroptosis, which provides new insights for treating OV patients with wild-type BRCA1/2 [7].

The proposal of ferroptosis has brought attention to the role of metal ions in cancer progression. Subsequently, Peter Tsvetkov et al. demonstrated a new type of cell death pathway prompted by copper named cuproptosis [8]. Copper binds directly to the lipoylated components of the tricarboxylic acid cycle (TCA) in mitochondrial respiration, resulting in an abnormal accumulation of lipoylated proteins. Protein-toxic stress responses are produced, which eventually lead to cell death. Currently, chemotherapy is still a chief treatment for OV, and a combination of platinum and paclitaxel remains a primary chemotherapy regimen; interestingly, previous reports have corroborated that copper transporters have a unique contribution to platinum resistance. Ctr1, the primary copper inflow transporter, can transport cisplatin, and the two external copper transporters, ATP7A and ATP7B regulate cisplatin’s outflow [9,10].

In summary, both cuproptosis and ferroptosis play crucial roles in OV [11,12]. However, whether cuproptosis combined with ferroptosis-related genes can be used as a marker of prognosis, drug sensitivity, and immunotherapy response in OV has yet to be elucidated. 

Constructing molecular subtypes and exploring the characterization of the corresponding immune microenvironment is extremely necessary for predicting the prognosis and selecting treatment regimens. Therefore, this study combined genes associated with cuproptosis and ferroptosis to explore their expression and roles in OV comprehensively. Specific molecular subtypes were identified for prognosis prediction and drug response assessment. We hope our study will guide the precise treatment of OV.

## 2. Materials and Methods

### 2.1. Data Processing

We downloaded RNASeq data and relevant clinical details from the UCSC Xena, Genotype-Tissue Expression Project (GTEx), GDC Data Portal, and GEO databases. We converted and normalized the FPKM values of TCGA-OV (376 OV patients) and GTEx-ovary (88 normal samples) to TPM values and finally combined GSE26712 data (185 OV patients) to obtain an integrated expression matrix. The clinical information of the patients included in this experiment is shown in Appendix A. Moreover, we adopted 16 cuproptosis-related genes (*CRGs*) and 60 ferroptosis-related genes (*FRGs*) based on previous reports [13,14,15,16], listed in Appendix A. HALLMARK gene set was gained from the MSigDB. Tumor Immune Single Cell Hub [17] was applied to explore gene expression at single-cell level.

The flowchart of the study design was illustrated in Figure 1.

### 2.2. Tissue Collection

Cancer and normal tissues came from ovarian cancer patients from Tongji Hospital, Huazhong University of Science and Technology. This study was approved by the Ethical Committee of Tongji Medical College, Huazhong University of Science and Technology (2021-S136)

### 2.3. Cell Culture

Ovarian cancer cell lines C13K, OVCAR-8, HOC7, A2780, SKOV3, and ES2 were included in our study. C13K, OVCAR-8, A2780, and HOC7 were maintained in RPMI-1640. SKOV3 and ES2 were cultured in McCoy’5A. A humidified 5% CO_2_ incubator at 37 °C has been utilized to culture all cells.

### 2.4. Real Time PCR

The TRIzol reagent (Invitrogen, Thermo Fisher Scientific, Waltham, MA, USA) was applied to extract RNA, while cDNA was synthesized using HiScript III reverse transcriptase (Vazyme, Nanjing, China). The relative mRNA expression level was measured by the 2^−∆∆CT^ method. The primer sequences were FDX1-F: 5′-TTCAACCTGTCACCTCATCTTTG-3′, FDX1-R: 5′-TGCCAGATCGAGCATGTCATT-3′; GFPT2-F: 5′-ATGTGCGGAATCTTTGCCTAC-3′, GFPT2-R: 5′-ATCGAGAGCCTTGACTTTCCC-3′; VSIG4-F: 5′-ACCAAGACTGAGGCACCTAC-3′, VSIG4-R: 5′-TCCAAGGTAGCCATCCATGT-3′; HOXA5-F: 5′-AACTCATTTTGCGGTCGCTAT-3′, HOXA5-R: 5′-TCCCTGAATTGCTCGCTCAC-3′; GAPDH-F:5′-GTATGACAACAGCCTCAAGAT-3′, GAPDH-R: 5′-GTCCTTCCACGATACCAAAG-3′.

### 2.5. Protein Level Analysis

Western blot: After lysing cells with RIPA lysis buffer, we isolated the total protein on SDS-polyacrylamide gel and transferred it to PVDF membranes. Antibody was purchased from FDX1/ADX Rabbit mAb (#A20895, Abconal, Wuhan, China) and diluted in a ratio of 1:1000.

Immunohistochemical staining (IHC) and immunofluorescence (IF): After cutting the sample tissues immersed in formalin to a thickness of 5 μm, we placed them on glass slides. Subsequently, we blocked the tissues in goat serum for one hour with endogenous peroxidase activity blocked. Tissues covered with the diluted anti-FDX1 at a ratio of 1:100 were stored at 4 °C overnight. A secondary antibody was manipulated to visualize.

### 2.6. Consensus Clustering and Immune Infiltration Evaluation 

First, based on the TCGA-OV and GTEx-ovary, we screened out 16 differentially expressed *CRGs* (Figure 2A) and 57 differentially expressed *FRGs* via the “limma” package (Appendix A). Then the effect of 57 *FRGs* and 16 *CRGs* was evaluated on OV prognosis through Kaplan-Meier analysis performed by the “survival” and “survminer” R package. Finally, 39 prognosis-related genes (*PRGs*) were identified for our subsequent analysis (Appendix A).

The “ConsensusClusterPlus” package was used to distinguish the individuals into exceptional subtypes according to the expression of *PRGs*. Similarly, KM analysis was employed to examine curves of OS in different subtypes. Following this, we conducted GSVA functional enrichment assay in subtypes based on the kegg.v7.4.symbols.gmt. The “clusterProfiler” package was used to complete statistical analysis [18]. Furthermore, using single-sample gene set enrichment analysis (ssGSEA), we examined immune cell infiltration and TME differences in two subtypes. ESTIMATE algorithm [19] was adopted to calculate tumor purity based on immune and stromal cell ratios for transcriptome expression data. Next, we explored the sensitivity of subtypes to drug treatments according to the Genomics of Drug Sensitivity in Cancer (GDSC) database [20]. Paclitaxel, Gemcitabine, Etoposide, Docetaxel, Lapatinib, Doxorubicin, and Cisplatin were considered in our study.

### 2.7. Identification of DEGs and Functional Enrichment Analysis

The “limma” package was used to identify the differential expression genes (*DEGs*) between different subtypes with an adj. *p* < 0.05 and |log2FC| ≥ 0.585. The “clusterProfiler” package was adopted for enrichment analysis to further complement the gene function associated with *DEGs*.

### 2.8. Risk Model Construction

Prognosis-associated *DEGs* were confirmed by univariate Cox analysis. For a more comprehensive evaluation, the second Consensus Clustering was computed based on these prognosis-related *DEGs* to separate individuals into specific gene clusters. As mentioned earlier, we thoroughly analyzed the differences between the immune microenvironments of different clusters. At the same time, principal component analysis (PCA) was applied to further compare the composition of different clusters. Ultimately, the TCIA database was scanned to evaluate the response based on TCGA-OV. 

We then attempted to construct a cuproptosis- and ferroptosis-related signature of prognosis. First, a 1:1 ratio was considered to assign all individuals to train and test groups randomly (Appendix A). Based on the training group, a prognostic COX model was identified, and the test group was utilized to confirm the model’s efficacy. To minimize the possibility of overfitting the model, the least absolute shrinkage and selection operator (LASSO) was computed. The trajectory of change in each independent variable was analyzed and significant genes were obtained using 10× cross-validation to establish risk signature. Formula: risk score = Σ (gene Exp × coefi) was fulfilled using Cox regression analysis to compute the risk score. Individuals were categorized into different risk groups according to the median risk score of the training group, and ROC curves and Kaplan–Meier analysis were evaluated. The R “glment” package was executed.

### 2.9. Risk Signature Validation

A “heatmap” package was adopted to demonstrate differences between two risk cohorts expressing prognosis-related *PRGs*. Next, to evaluate the implication of clinical features and risk score on overall survival (OS), univariate and multivariate Cox analysis were conducted. In addition, we performed the “RMS” package to plot a nomogram containing clinical information and risk score for predicting the OS, and the nomogram effectiveness was evaluated using a calibration curve.

### 2.10. Association of Risk Model with Immune-Related Factors

The differences between immune microenvironments of the high-risk and low-risk groups were compared by ssGSEA analysis. PCA and tSNE analysis were applied to gain a thorough result. The “ggalluvial” package was used to draw the association among different PRGclusters, gene clusters, risk layers, and fustat status. Next, the “estimate” package was carried out to assess the differences between immune checkpoint gene (ICP) expression and the immune microenvironment in the risk^high^ and risk^low^ groups. The association between prognosis-related hub genes and immune cells was similarly explored. For Gene Set Enrichment Analysis (GSEA), GSEA_4.2.3 was used (University of California San Diego, La Jolla, CA, USA and Broad Institute, Cambridgeu, MA, USA). Somatic mutation profiles in patients with different risk layers were done using the maftools program. 

### 2.11. Drug and Immunotherapy Potential Detection in the Risk Model

The “pRRophetic” package and TCIA database were applied to detect the potential drugs for individual treatment. Besides, the immunotherapy value of the model was validated in the IMvigor210 cohort (a public dataset of immunotherapy for urinary urothelial tumors) [21] and GSE35640 (immunotherapy dataset for metastatic melanoma and non-small cell lung cancer).

### 2.12. Statistical Analysis

Pearson’s correlation coefficient test and two-tailed Student’s *t*-test were adopted to conduct expression analysis. HR and *p*-value were used as indicators of survival. Moreover, data integration and processing were completed in R 4.1.2 (R Foundation for Statistical Computing, Vienna, Austria) and R studio.

## 3. Results

### 3.1. Expression Landscape of 16 CRGs in Ovarian Cancer

We first evaluated the expression profile of 16 *CRGs* in ovarian cancer combined TCGA-OV with GTEx-ovary samples. In the tumor, a heatmap revealed that DLD, PHDB, ATP7B, SLC31A1, CDKN2A, FDX1, DLAT, ATP7A, MTF1, and DBT expression upregulated, despite LIAS, LIPT1, GCSH, DLST, PDHA1, and GLS yielding the contrary trend (Figure 2A).

Additionally, based on GSE154600, we analyzed the expression of 16 *CRGs* at the cellular level (Figure 2B).

FDX1, the dominant gene of cuproptosis, plays an essential role in cuproptosis. We then verified the expression of FDX1 in ovarian cancer cells at the transcription and protein levels (Figure 2C,D). The mRNA level was inconsistent with protein expression in some cell lines, possibly due to post-transcriptional protein modifications. IHC results implied that, in contrast to normal tissues, FDX1 expression in malignant tissues appeared to be higher (Figure 2E). Figure 2F depicts the expression of FDX1 in OV patients.

### 3.2. Molecular Subtypes Identification for Prognosis Prediction

As depicted above, we finalized the identification of 39 genes for cuproptosis and ferroptosis associated with prognosis for our subsequent analysis, including 7 *CRGs* and 32 *FRGs*, and Figure 3A demonstrates their interaction. 

Based on the expression of 39 *PRGs*, the individuals were divided into molecular subtypes A (*n* = 327) and B (*n* = 234) through cluster analysis (Figure 3B, Appendix A). The OS of individuals in the two subtypes differed significantly. Individuals in subtype B had shorter OS (Figure 3C). Heatmap implied the expression discrepancy of 39 *PRGs* and their links to clinically relevant information (Figure 3D). We also explored the differences in the expression of ICP genes between individuals with two subtypes, and we discovered that subtype B was positively correlated with ICP (Figure 3E). In the GSVA analysis, we found that subtype B was associated with more tumor-associated signal pathways, such as focal adhesion and cell adhesion molecules cams pathways (Figure 3F). It probably explained the poor prognosis for patients despite having higher levels of immune infiltration in subtype B (Figure 3G,H). Similarly, compared with subtype A, a higher stromal score was exhibited in subtype B. 

At the same time, in drug susceptibility analysis, different subgroups responded differently to chemotherapy drugs. Of the seven commonly used medications for OV that we included, individuals in subtype B were sensitive to Doxorubicin and Gemcitabine, while individuals in subtype A were more sensitive to Etoposide and Docetaxel. Unfortunately, the two subtypes implied no significant difference in sensitivity to Cisplatin and Paclitaxel data now shown, the first-line chemotherapy drugs for OV (Figure 3I).

### 3.3. Secondary Clustering: Prognostic-Related DEGs Identification; Quantifying Cuproptosis and Ferroptosis Patterns Substantiate Strong Support for Predicting Prognosis

First, we obtained 322 *DEGs* based on the gene expression between two subtypes (Figure 4A). Subsequently, GO enrichment analysis was scanned to explore further the gene function associated with *DEGs*. The results implied *DEGs* were intimately related to extracellular matrix organization (Figure 4B).

We then strived to search prognosis-related *DEGs* by applying univariate Cox regression analysis and eventually screened out 152 genes associated with prognosis. Subsequently, according to the expression of the 152 genes, we assigned individuals to gene cluster A (*n* = 298) and B (*n* = 263) using Consensus Clustering analysis (Figure 4C, Appendix A). And considerable differences existed between the two groups in the PCA model (Figure 4D). Meanwhile, the gene cluster presented an excellent predictive value for individual survival. In the KM analysis of OS, the survival time of group B was significantly extended. (Figure 4E). 

Like PRGcluster, functional analysis outcomes implied group A was positively connected with tumor-related pathways, such as KEGG-pathways-in-cancer and KEGG-cell-adhesion-molecules-cams (Figure 4F). Of note, while group A individuals have higher levels of immune infiltration, more matrix components in the immune microenvironment were occupied (Figure 4G,H). Similarly, in drug susceptibility analysis, we found differences in the sensitivity of individuals in the two clusters to chemotherapy drugs. Group B patients were more likely to benefit from Etoposide and Paclitaxel.

For the A cluster, Gemcitabine and Docetaxel were better options (Figure 5A). Furthermore, many chemotherapy-resistant patients have demonstrated immune checkpoint block (ICB) efficacy against PD-1/PD-L1 and CTLA4 [22,23]. Reassuringly, cluster A presented a better therapeutic response (Figure 5B).

### 3.4. Prognostic Risk Model Establishment

We randomly divided individuals into train (*n* = 264) and test (*n* = 264) groups on average. Among them, the training group was utilized for subsequent analysis. We applied LASSO regression to analyze the 152 *DEGs* checked above. As illustrated in Appendix A, the trajectory of change in each independent variable was analyzed. After cross-validation, we identified nine genes (*GFPT2*, *OLFML3*, *VSIG4*, *ADH1B*, *STAB1*, *ID1*, *HOXA5*, *CXCL9*, and *LYPD1*) for the later multivariate Cox regression analysis. In the end, five genes were recognized as significant prognostic factors and measured in constructing the predictive risk model. The risk score was computed based on the formula: Risk score = (0.2720 × GFPT2 expression) + (0.2125 × VSIG4 expression) + (0.1053 × HOXA5 expression) + (–0.2408 × CXCL9 expression) + (–0.0705 × LYPD1 expression) (Figure 5C–E). 

### 3.5. The Risk Model Performs A Substantial Predictive Prognostic Value

For the entire set, through KM analysis, it was observed that OS was considerably prolonged in the low-risk group (Figure 5F). According to the ROC curve, the prognostic model possessed a satisfying potential to predict 1, 3, and 5-year OS values (Figure 5G). The results of PCA and tSNE analysis for the high- and low-risk groups are displayed in Figure 5H,I. Specific details of the train and test sets were displayed in Appendix A. Figure 5J was drafted using the “ggalluvial” package to visually analyze the connections between PRGclusters, gene clusters, risk layers, and survival status. Risk scores differed significantly between the two PRGclusters and gene clusters (Figure 5K). Patients with PRGcluster B and gene cluster A had significantly higher risk scores.

Notably, with univariate and multivariate Cox analysis, we determined a unique contribution of risk scores for predicting OS (Figure 6A, all *p* < 0.001). In addition, as illustrated in Figure 6B, we defined a nomogram to predict the OS better. The closer the calibration curve is to the diagonal, the better the predictive performance of the nomogram (Figure 6C). Subsequently, we assessed the clinical characteristics of patients in different risk groups (Figure 6D).

### 3.6. Immune Infiltration Abundance in the Risk Model

First, we analyzed the link between five dominant genes used to build the model and immune infiltration. We found that CXCL9 regulated a positive role in immunity; as shown in Figure 6E, CXCL9 was positively correlated with M1, CD8+T cells, dendritic cells (DC), CD4+T cells, and Th cells. In contrast, VSIG4 and GFPT2 were associated with M2 and negatively correlated with activated NK cells. 

Tumor-associated macrophages (TAMs) are involved in the formation of the TME. They are widely present in various tumors to promote tumor growth, invasion, metastasis, and drug resistance. M1 maintains an anti-tumor effect, while M2 promotes cancer proliferation and aggression, and the two forms are in a state of continuous transition [24,25]. Through the CIBERSORT algorithm, we noticed an inverse connection between risk score with macrophage M1 and trended positively with M2. Similarly, the risk score is negatively connected with CD8+T cells, CD4+T cells, and NK cells (Figure 6F, Appendix A).

Furthermore, we probed that infiltration of immune cells was more significant in the low-risk group, including CD8+T cells, NK cells, DC cells, Th1, and Th2 cells. A similar phenomenon also appeared in the analysis of immune function; abundant, powerful anti-tumor immune functions were generated in the low-risk cohort (Figure 7A,B). 

HLA genes control mutual recognition between cells and regulate the immune response. In addition, prior investigations have queried that the abnormal expression of HLA genes is highly associated with OV. For example, one of the mechanisms of ovarian cancer evading immune surveillance is to upregulate human leukocyte antigen-G (HLA-G) expression [26,27]. Hence, we explored the expression of HLA genes in different risk layers to help treat OV. Results showed that individuals with low scores showed a notable increase in HLA gene expression (Figure 7C). Moreover, the expression of ICP genes was remarkably upregulated in the low-risk cohort (Figure 7D). 

### 3.7. Benefits from Drugs and Immunotherapy

In our experiment, IMvigor210 and GSE35640 data were referenced for validating the correlation between risk scores and ICB responses, and we were pleased to find that immunotherapy was more effective in the low-risk cohort. (Figure 7E). Additionally, individuals with low-risk scores responded stronger to receiving anti-PD1, anti-CTLA4, or anti-PD1 and anti-CTLA4 combination therapy (Figure 7F). The drug sensitivity analysis strengthens that, except for Docetaxel, chemotherapy drugs were more sensitive to individuals in the low-risk group (Figure 7G). 

### 3.8. Somatic Mutation, TMB, Stemness Scores Analysis

Figure 8A,B shows that low-risk individuals owned a higher somatic mutation frequency (97.46% vs. 94.74%). Additionally, TP53 is the most commonly mutated gene, and TP53 mutations are present in at least 80% of patients with OV. Especially in high-grade serous ovarian cancer, about 96% of patients are reported to occur TP53 mutations [28,29]. In our model, TP53 occupied the highest mutation frequency in both groups, 90% in the low-risk cohort and 83% in the high-risk set. 

Likewise, predicting the efficacy of ICB still needs a definitive biomarker. TMB, consistent with PD-L1 expression, is a proxy for neoantigen burden. Clinical studies have exhibited that patients with lower TMB levels have more prolonged survival and higher response rates after ICB treatment [30]. In our study, although the difference in TMB was not prominent, we still observed that individuals with low scores and high TMB retained a longer OS (Figure 8C–F). Furthermore, a significant negative correlation was presented in the correlation analysis of the risk score with tumor stemness scores (Figure 8F, R = 0.39, *p* < 0.001).

Subsequently, we found that the high-risk cohort enriched multiple tumor-associated pathways through the KEGG pathway analysis. In contrast, the low-risk cohort enriched more immune-related pathways via GO-BP analysis (Figure 8G,H).

In summary, individuals with low scores had a higher degree of immune infiltration, a stronger response to drugs and immunotherapy, and a longer survival time.

### 3.9. Expression Verification of Five Hub Genes at the Cell Level

As a supplement, we validated the expression of five dominant genes at the cell level relying on GSE154600. As Figure 8I depicted, GFPT2 was mainly expressed in fibroblasts, VSIG4 and CXCL9 were primarily expressed in monocytes, and LYPD1 was expressed in malignant epithelial tissues. As for HOXA5, it was weakly expressed on both fibroblasts and monocytes. Finally, we selected three genes in the model and verified their expression in ovarian cancer cell lines (Figure 8J).

## 4. Discussion

Ovarian cancer is the deadliest gynecological malignancy with an alarmingly poor prognosis due to late detection and chemotherapy resistance. The concept of molecular typing provides a new basis for exploring heterogeneity, prognostic assessment, and individualized treatment options in OV. Therefore, we urgently need to construct molecular subtypes to provide more personalized treatment options and prolong OS.

Although numerous studies have shown the importance of cuproptosis and ferroptosis in OV, there are currently no studies that combine the two for a more comprehensive analysis. Here, 76 *CRGs* and *FRGs* were included in this study. We aggregated data from GTEx-ovary, TCGA-OV, and GSE26712 for systematic analysis. The prognosis-related subtypes were established based on the expression of 39 *PRGs* selected from 76 *FRGs* and *CRGs*. The prognosis and TME of individuals varied significantly between subtypes. Simultaneously, OV individuals were divided into gene clusters for more thorough consideration. Depending on the molecular subtypes, the prognosis and drug sensitivity of individuals could be better predicted.

Furthermore, LASSO and Cox regression analysis identified a risk model centered on GFPT2, VSIG4, HOXA5, CXCL9, and LYPD1. Patients were split into different risk layers. Remarkably, individuals with lower risk scores displayed a stronger correlation with immune-related factors and made it easier to benefit from chemotherapy and immunotherapy, explaining why the low-risk group presented a more satisfying outcome than the high-risk group. Moreover, ICP genes expressed drastically enhanced, including PD-L1 and PD-1 in the low-risk group. Conversely, the high-risk group was associated with more carcinogenic mechanisms via GSEA_4.2.3.

In the past few years, molecularly targeted therapies and immunotherapies have led to longer-term survival for OV patients. For example, as a key signaling factor affecting angiogenesis, vascular endothelial growth factor receptor (VEGFR) has become an important target against tumors [31]. However, primary and acquired resistance is also emerging. In recent years, the immunosuppressive effect of VEGF has been gradually elucidated [32]. The study has reported increased Tregs infiltration and enhanced expression of PD-L1 in tumor tissues after receiving antiangiogenic therapy [33]. The functional inhibition of T cells and the inhibitory effect of myeloid-derived suppressor cells (MDSCs), mast cells, and DCs deserve our attention, so the combination therapy of VEGFR inhibitors and immunotherapy is a research direction worth exploring [34]. Moreover, clinical studies of anti-PD-1/PD-L1 also suggest that high PD-L1 expression appears more responsive in OV [35]. In our study, differences in PD-L1 expression and immune status of individuals in different molecular types showed good differentiation, which has relatively far-reaching implications.

However, some limitations remain in this study. First, only 76 *CRGs* and FRGs were considered. But in clinical trial design and enrolments, we should think more about genome-directed stratification. Currently, genetic testing is recommended for front-line treatment in OV, including BRCA1/2, HRD, PALB2, BARD1, etc. Studies have proved that higher mutational burden and PD-L1 expression in ovarian cancer are associated with BRCA mutations [36], which may make these cancers more suitable for ICB therapy [37]. Additionally, wild-type BRCA ovarian cancer is known to be more aggressive and has a worse prognosis, especially in HRD-negative patients. Therefore, in the future, we will fully consider the BRCA status of patients and try to provide more effective personalized treatment for patients. In a word, more consideration should be taken in the future. Second, this study solely pertains to conducting superficial experimental verification, and we need more experiments to verify the biological function of the five core genes in the model. Third, the correlation between cuproptosis and the carcinogenic mechanism of OV deserves further exploration, which in turn provides a new approach to treating OV. 

## 5. Conclusions

Our study is the first to combine *CRGs* with *FRGs* to explore their roles in OV. We performed cluster analyses twice to confirm different molecular types based on the genes associated with cuproptosis and ferroptosis. A risk model of five dominant genes was established through Lasso analysis. The results show that our molecular subtypes and risk model present potential prognosis and immune prediction ability. This study aimed to provide a reference for the individualized treatment of patients with ovarian cancer.

## Figures and Tables

**Figure 1 cancers-15-00579-f001:**
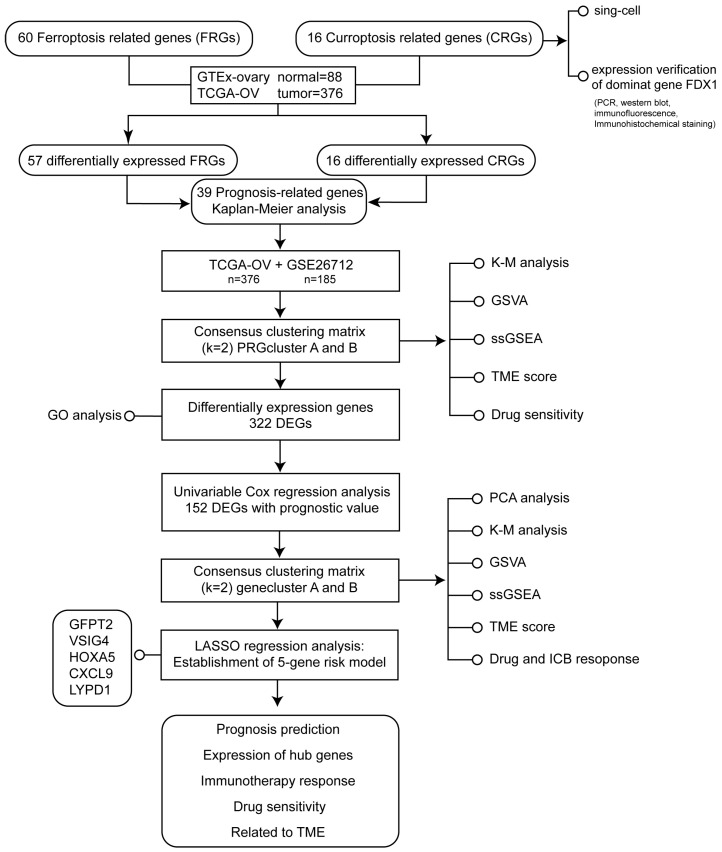
The flowchart of the study design.

**Figure 2 cancers-15-00579-f002:**
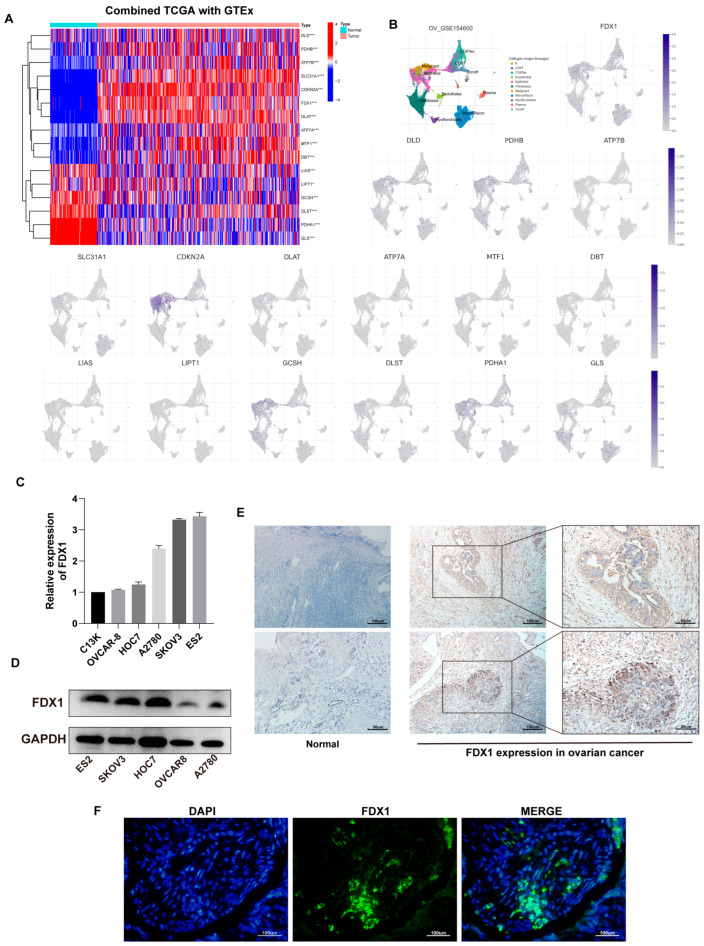
Expression of *CRGs* in OV. (**A**) Differential profile of expression of 16 *CRGs* in TCGA-OV and GTEx-ovary. (**B**) Expression of 16 *CRGs* at the cellular level based on GSE156400. (**C**) FDX1 expression in ovarian cell lines at the transcriptional and (**D**) protein levels. Immunohistochemical staining (**E**) and immunofluorescence (**F**) results of FDX1 protein in OV and normal samples. Original blots see Appendix A.

**Figure 3 cancers-15-00579-f003:**
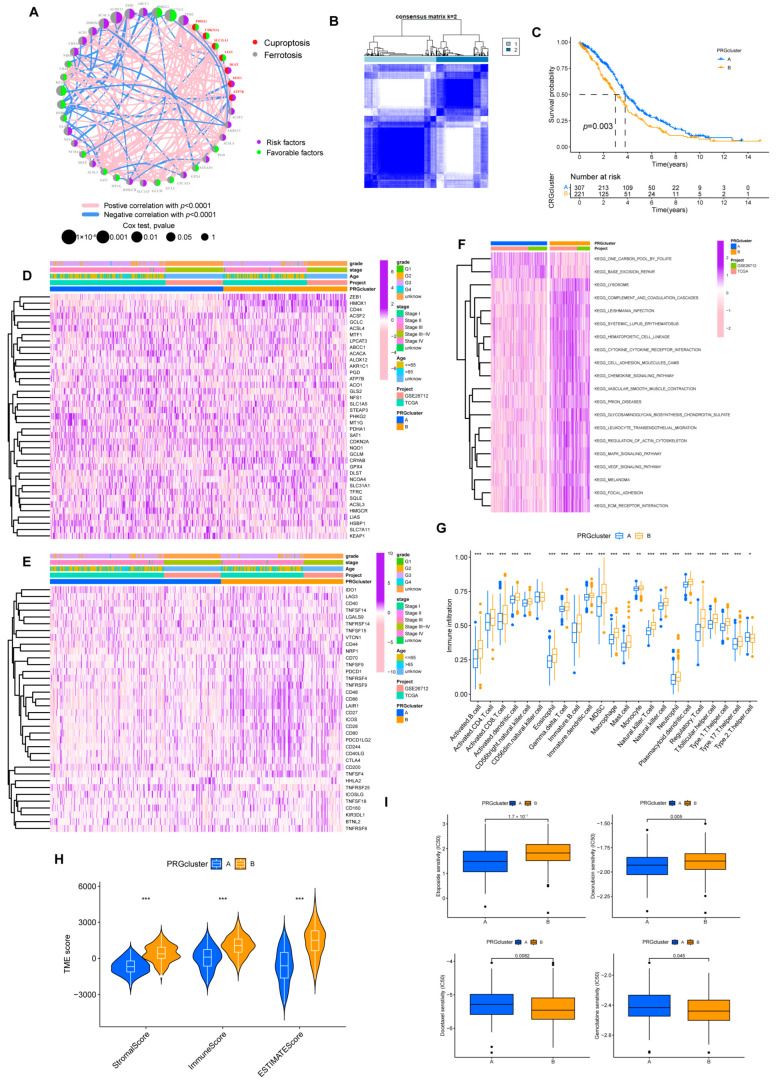
Prognosis subtypes identification. (**A**) Network diagram of 39 *PRGs*. (**B**) Consensus matrixes for all samples were clustered into specific subtypes (k = 2). (**C**) Prognostic differences between two subtypes. (**D**) Heatmap of the correlation of subtypes with clinical features. (**E**) Heatmap of the correlation of subtypes with ICPs. (**F**) GSVA analysis of two subtypes. (**G**) Overview of immune infiltration between the two subgroups. (* *p* < 0.05; ** *p* < 0.01; *** *p* < 0.001). (**H**) Different subtypes differ in immune microenvironment scores (stromal and immune scores). (*** *p* < 0.001). (**I**) Sensitivity of subtypes to the chemotherapeutic drugs.

**Figure 4 cancers-15-00579-f004:**
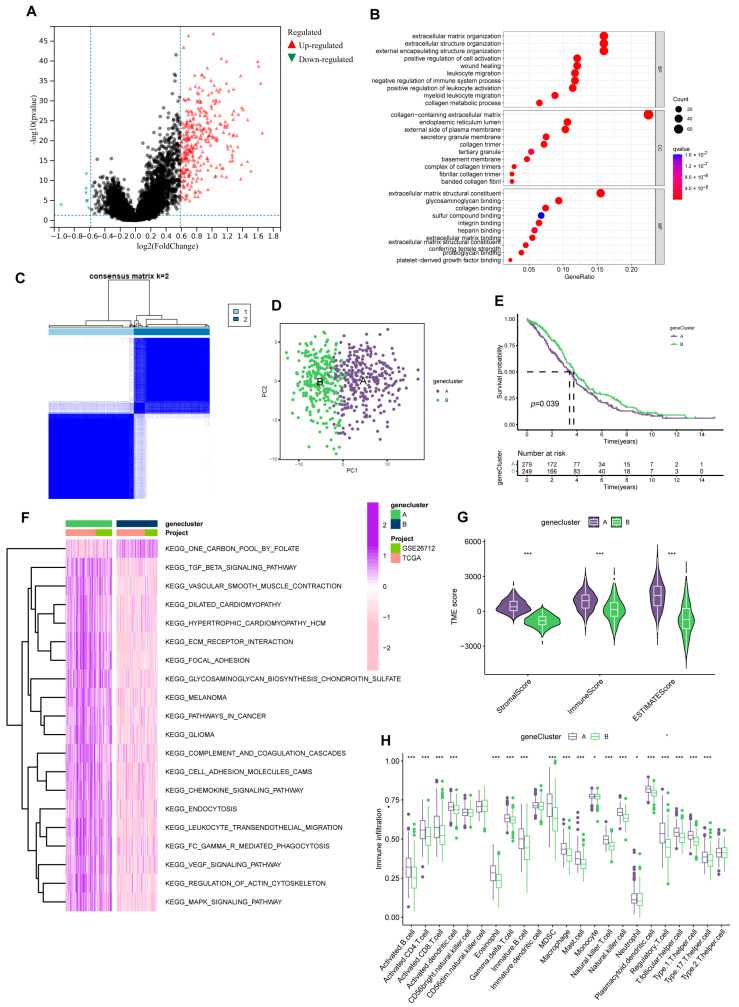
Prognosis-related gene clusters identification. (**A**) The *DEGs* of subtype A and subtype B. (**B**) GO analysis of 322 *DEGs*. (**C**) The consensus matrix identifies two gene clusters (k = 2). (**D**) PCA analysis demonstrated significant transcriptome differences between the two clusters. (**E**) Gene clusters were closely related to OS in OV. (**F**) GSVA analysis of the KEGG pathways in gene clusters. (**G**) Different clusters differ in TME scores (stromal score and immune score). (*** *p* < 0.001). (**H**) Immune function analysis of two sets. (* *p* < 0.05; *** *p* < 0.001).

**Figure 5 cancers-15-00579-f005:**
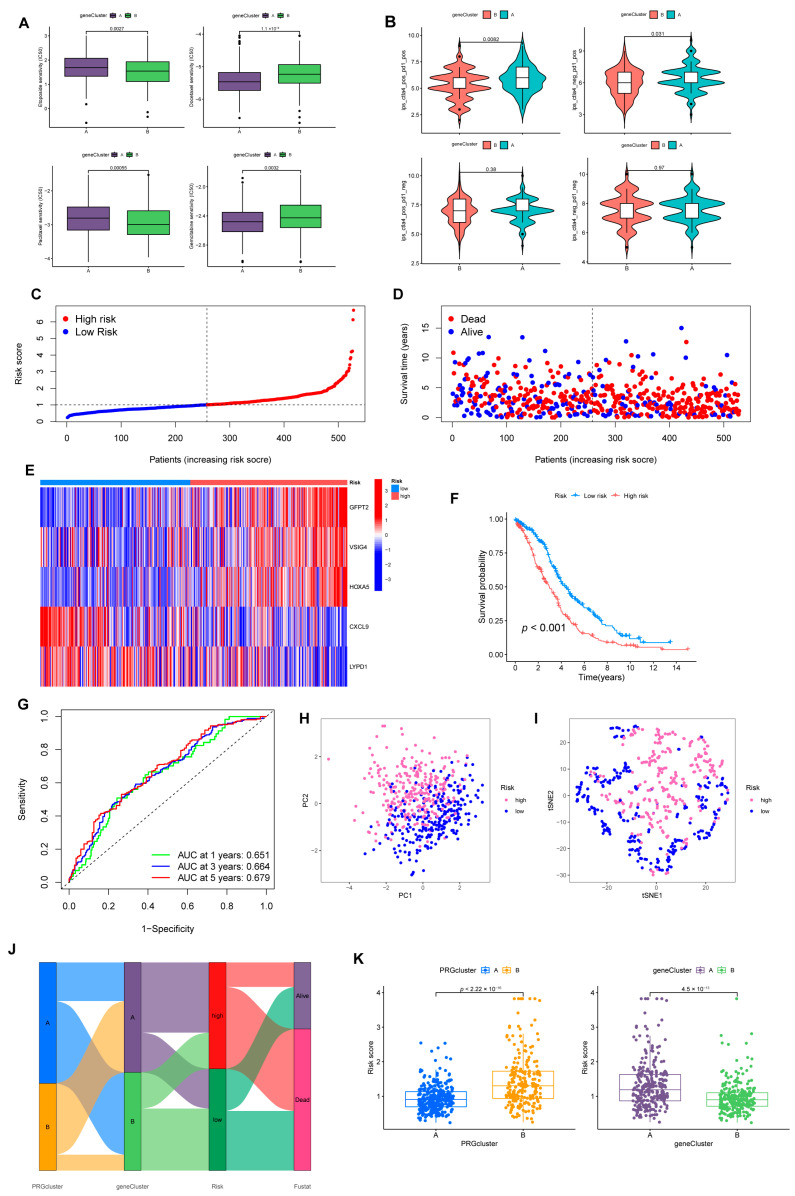
Construction of risk model associated with *CRGs* and *FRGs*. (**A**) Sensitivity of two gene-clusters to the chemotherapeutic drug commonly targeted OV. (**B**) Response of different clusters to immune checkpoint inhibitors. (**C**) Exhibition of a risk model in the entire cohort. (**D**) Overview of the survival time and status of the entire cohort between the two risk groups. (**E**) A heatmap of 5 hub genes expression in the entire cohort. (**F**) Differences in OS between different risk groups in the entire set. (**G**) ROC analysis predicts OS at 1, 3, and 5 years according to the risk score in the entire set. (**H**) PCA and (**I**) tSNE analysis results for different risk layers in the entire set. (**J**) Association with PRGcluster, gene cluster, risk score, and survival outcome. (**K**) Distribution profile of risk scores in different PRGsubtypes and gene clusters.

**Figure 6 cancers-15-00579-f006:**
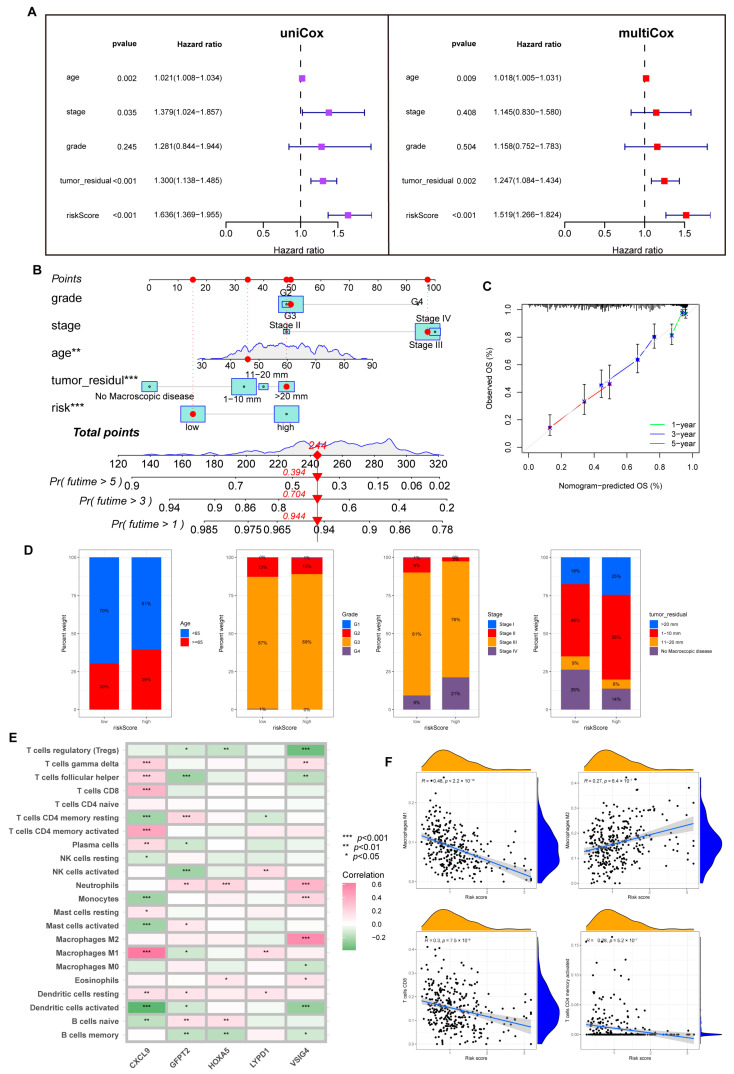
The predictive value of the risk model. (**A**) Uni-Cox and multi-Cox analysis between OS and related factors. (**B**) A nomogram was established for OS prediction. (** *p* < 0.01; *** *p* < 0.001). (**C**) Calibration curve for evaluating nomogram performance. (**D**) Overview of clinical information for different risk individuals. (**E**) Correlation of five hub genes and immune cell infiltration. (**F**) Risk score correlation of immune infiltration abundance.

**Figure 7 cancers-15-00579-f007:**
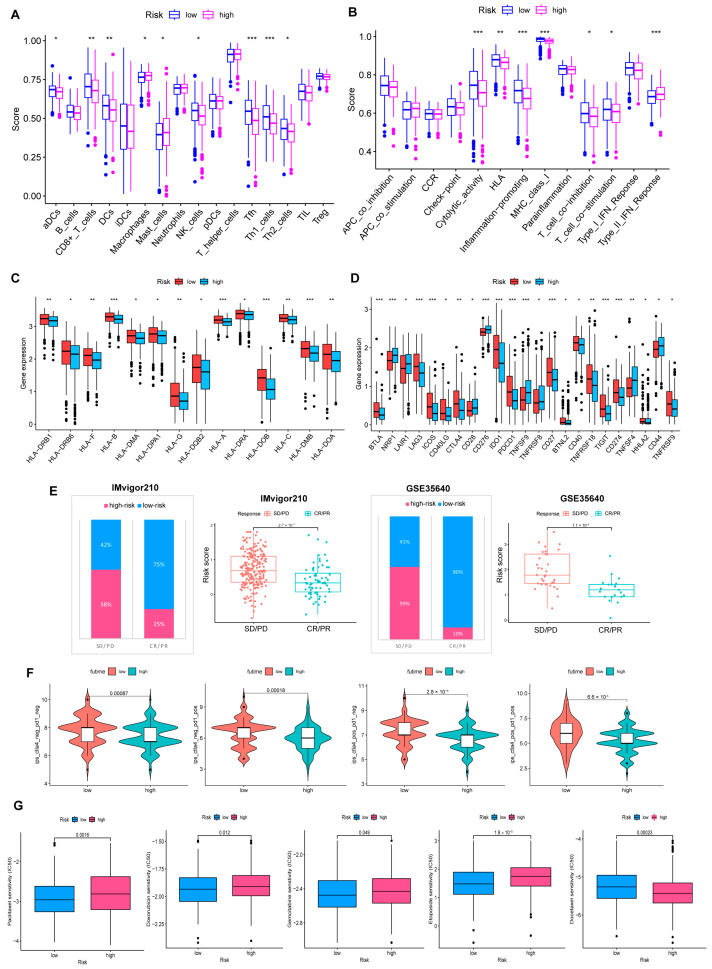
Comprehensive analysis of the TME of risk cohorts. (**A**,**B**) Differences in immune cell infiltration and immune function between two risk groups. (**C**) The two groups have differences in immune HLA gene expression and (**D**) ICP gene expression. (* *p* < 0.05; ** *p* < 0.01; *** *p* < 0.001). (**E**) Validate the predictive value of the model in the immunotherapy group. (**F**) Immunotherapy responsiveness of the two groups. (**G**) Association between risk score and susceptibility to chemotherapy.

**Figure 8 cancers-15-00579-f008:**
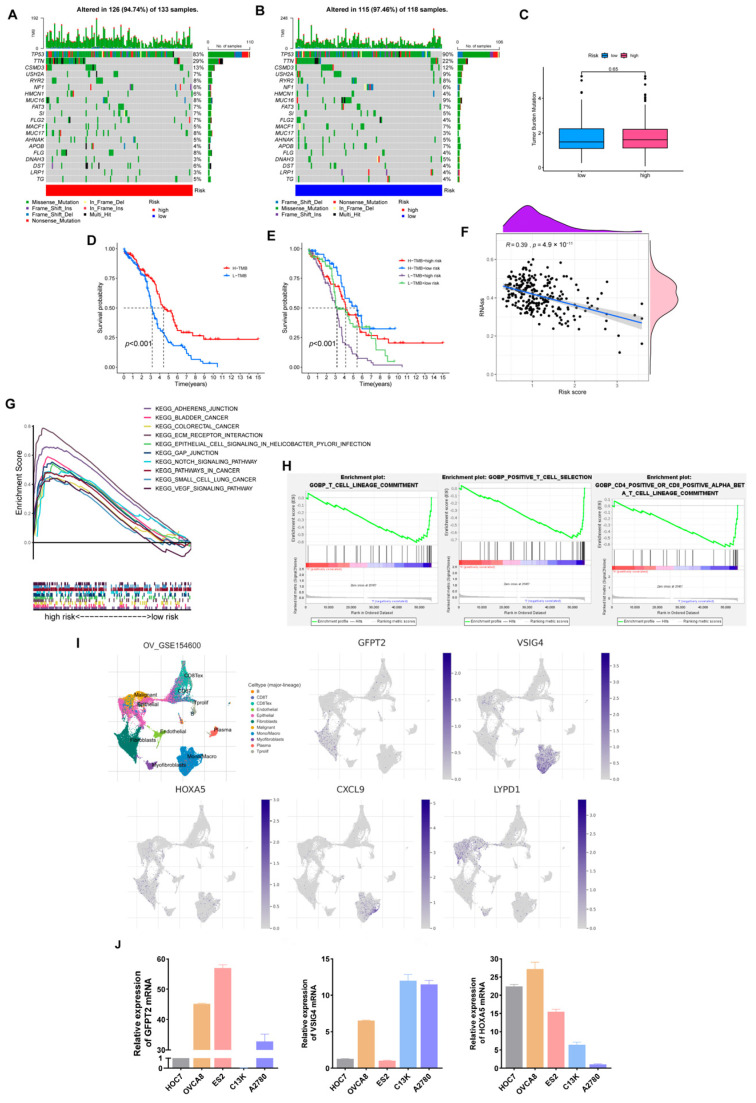
Comprehensive analysis of the risk model in OV. (**A**,**B**) Somatic mutation landscape in different risk sets. (**C**) TMB comparison of two groups. (**D**,**E**) Relevance of TMB and risk scores to prognosis. (**F**) Correlation of risk scores and Stemness Scores. (**G**) KEGG pathways enriched in the high-risk set. (**H**) GO-BP enrichment results in the low-risk cohort. (**I**) The expression of five hub genes is based on GSE154600. (**J**) Verification of three hub genes’ expression in ovarian cancer cell lines.

## Data Availability

The data applied to support the results of this study are available at the UCSC Xena database https://xena.ucsc.edu/ (accessed on 28 April 2022), GDC Data Portal https://portal.gdc.cancer.gov/ (accessed on 28 April 2022), Genotype-Tissue Expression Project (GTEx) https://www.gtexportal.org/ (accessed on 27 May 2022), GEO database https://www.ncbi.nlm.nih.gov/geo/ (accessed on 27 June 2022), and TCIA database https://www.tcia.at/ (accessed on 30 June 2022). Expression analysis was detected by TISCH http://tisch.comp-genomics.org/ (accessed on 2 October 2022).

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
