# Peer review of "Identification of a Novel Model for Predicting the Prognosis and Immune Response Based on Genes Related to Cuproptosis and Ferroptosis in Ovarian Cancer"

_cancers, 2023, doi:10.3390/cancers15030579_

Round 1

Reviewer 1 Report

Ying Li et al. identified a potential novel model for predicting the prognosis and immune response based on genes related to cuproptosis and ferroptosis in ovarian cancer. 

Points to be addressed:

1) The rationale of why the authors came up with this research is scanty and is related to a lack of novelty: please highlight what this manuscript might add.

2) What is the information that is not exactly available that motivated the authors to come up with this information. What are the current caveats and how do the authors highlight the current research in answering them? If not they need to address in background and infuture directions .

3)State of the art figures are required: scale bar should be provided in high resolution.

4)The authors could provide a little more consideration of genomic directed stratifications in clinical trial design and enrolments. 

5)The underlying message here is that more precision and individualized approaches need to be tested in well-designed clinical trials – a challenge, but I would be interested in their perspective of how this might be done. If beyond the scope of the manuscript, this should be highlighted as a limitation

6) The authors need to highlight what new information the review is providing to enhance the research in progress

7) Did the author check for gaussian dinstribution of the data?

9) this reviewer personally misses some insights regarding angiogenesis and immune patrolling:As is now well known, tumors grow and evolve through a constant crosstalk with the surrounding microenvironment, and emerging evidence indicates that angiogenesis and immunosuppression frequently occur simultaneously in response to this crosstalk. Accordingly, strategies combining anti-angiogenic therapy and immunotherapy seem to have the potential to tip the balance of the tumor microenvironment and improve treatment response. Please refer to PMID: 34298648 and expand

Reviewer 2 Report

In this manuscript, Li et al., developed a cuproptosis and ferroptosis related gene model and performed systematic validation and exploration of various aspects, which is expected to predict the prognosis and immune response of ovarian cancer. Rational methods and statistical analysis were applied, and the findings showed certain significance and novelty. However, there are some flaws and issues that need further clarification.
Major points:

1. The manuscript is not well written, with a lot of typographical and grammar mistakes. The whole flow is really confused and hard to follow.

2. Please include a flowchart of this study.

3. What is the significance to start this study with a “Pan-cancer analysis of CRG”? The key findings of this research, including as the molecular subtypes and risk model, were not established based on these 10 CRGs.

If you want to include this results (Results 3.1 and 3.2) in the manuscript, please provide the necessity and importance of them to the whole study. Explanations should be given to the relationship between these 10 CRGs and the 16 CRGs for the following analysis. Results 3.2 should have a different sub-title, since it’s sharing the same one with 3.3. In results 3.2, if you want to prove that FDX1 is higher expressed in ovarian cancer cell lines, then you need to have normal ovarian cells as the control.

4. Please include the detailed clinical characteristics of the OC patients in a supplementary table. 

5. The methods how they running the analysis and judging the results were not sufficiently described, which is difficult to be repeated by other researchers.

Minor points:

1. Line 15: “molecular types was established based on 76 genes”. Should it be 76 or 39?

2. Line 194: please describe how you identify 39 PRGs from 16 CRGs and 60 FRGs in detail.

3. Line 197: How many individuals are there in each molecular subtype?

4. Line 213: Please add “data now shown” after “no significant difference in sensitivity to Cisplatin and Paclitaxel”.

5. Results 3.4 and 3.5 could be combined into one section. Fig. 4A is not well presented. It’s not necessary to include 4A in figure 4. If you want to keep it, please consider about the volcano plot.

Please provide explanations on the reason why a GO enrichment analysis is performed in Results 3.4?

6. Line 253: Supplementary Fig 3A is the result of LASSO analysis, but is not for group setting.

7. Line 264-265: Please figure out if Fig5. f.g are for the entire set or not.

8. Line 269: Please provide necessary explanations to Fig. 5K. Please confirm whether it’s “PRG” or “CRG” in Fig. 5K.

9. Please provide a legend for Fig. 7G.

10. The “Conclusions” need some improvement.

Round 2

Reviewer 1 Report

I am satisfied with the rebuttal provided.

Reviewer 2 Report

Thank you for the explanations and great efforts you have made. I believe that the manuscript has been sufficiently improved to warrant publication in this jouranl.